# Risk factors for hydrocephalus following fourth ventricle tumor surgery: A retrospective analysis of 121 patients

**Tengyun Chen**[☯], **Yanming Ren**[☯], **Chenghong Wang, Bowen Huang, Zhigang Lan, Wenke Liu, Yan Ju, Xuhui Hui, Yuekang Zhang**[ID]*

Department of Neurosurgery, West China Hospital, Sichuan University, Chengdu, Sichuan, PR China

☯ These authors contributed equally to this work.
* 2012zykyx@sina.cn

**Data Availability Statement:** All relevant data are within the paper and its Supporting Information files.

## Abstract

### Background and aim

Most patients who present with a fourth ventricle tumor have concurrent hydrocephalus, and some demonstrate persistent hydrocephalus after tumor resection. There is still no consensus on the management of hydrocephalus in patients with fourth ventricle tumor after surgery. The purpose of this study was to identify the factors that predispose to postoperative hydrocephalus and the need for a postoperative cerebrospinal fluid (CSF) diversion procedure.

### Materials and methods

We performed a retrospective analysis of patients who underwent surgery of the fourth ventricle tumor between January 2013 and December 2018 at the Department of Neurosurgery in West China Hospital of Sichuan University. The characteristics of patients and the tumor location, tumor size, tumor histology, and preventive external ventricular drainage (EVD) that were potentially correlated with CSF circulation were evaluated in univariate and multivariate analysis.

### Results

A total of 121 patients were enrolled in our study; 16 (12.9%) patients underwent postoperative CSF drainage. Univariate analysis revealed that superior extension (p = 0.004), preoperative hydrocephalus (p<0.001), and subtotal resection (p<0.001) were significantly associated with postoperative hydrocephalus. Multivariate analysis revealed that superior extension (p = 0.013; OR = 44.761; 95% CI 2.235–896.310) and subtotal resection (p = 0.005; OR = 0.087; 95% CI 0.016–0.473) were independent risk factors for postoperative hydrocephalus after resection of fourth ventricle tumor.

### Conclusion

Superior tumor extension (into the aqueduct) and failed total resection of tumor were identified as independent risk factors for postoperative hydrocephalus in patients with fourth ventricle tumor.

**Funding:** This work was supported by Program of Chengdu Science and Technology Bureau in the form of a grant awarded to YZ (2019-YF05-00392-SN). The funder had no role in study design, data collection and analysis, decision to publish, or preparation of the manuscript.

**Competing interests:** The authors have declared that no competing interests exist.

## Introduction

Posterior fossa tumors represent 7.9% of intracranial lesions and approximately 20–90% of patients with posterior fossa tumors present with hydrocephalus before tumor surgery.[1–6] Although in posterior fossa tumor, tumor removal can restore CSF circulation, 10–30% of patients tend to experience persistent hydrocephalus following posterior fossa resection.[4, 5, 7–9] The patients may present with signs and symptoms of increased intracranial pressure from hydrocephalus (headache, nausea/vomiting, vertigo, unsteady gait, diplopia, papilledema, etc.), which affect the patient's life quality and may result in a prolonged hospital stay.[2, 10] Fourth ventricle tumors, in particular, present a high rate of postoperative hydrocephalus because of the compression of tumor to the cerebrospinal fluid (CSF) pathways.[11, 12]

Surgery for fourth ventricular tumors is plagued by potential pitfalls caused by proximity to deep eloquent structures and the risk of injury to perforating arteries supplying subcortical regions and lesions of the fourth ventricle, which make up only a fraction of this subset.[13, 14] The lack of cases limits clinical experience data on the spectrum of pathologies in this region.[2–9, 15–18] We aimed to sought factors, which might be correlated with the development of persistent hydrocephalus following resection of fourth ventricle tumors to evaluate the indication for postoperative CSF drainage.

## Materials and methods

### Study population and data collection

This retrospective descriptive cohort study investigated the incidence of postoperative hydrocephalus and its causative factors in a consecutive group of patients who underwent surgery of fourth ventricle tumors between January 2013 and December 2018 at the Department of Neurosurgery in West China Hospital of Sichuan University.

The inclusion criteria were: 1) single intracranial neoplasm detected by preoperative magnetic resonance imaging; 2) surgical resection of the lesion; 3) a tumor confirmed by pathologic diagnosis. Patients were excluded if 1) there was evidence of multiple tumors or repeated surgery for tumor; 2) the patients underwent biopsy rather than resection.

The West China Hospital Ethics Committee approved this study. Individual patient identification data were not collected when the database was developed, and this is the reason why patient consent was not required for this study.

After selecting the patient population, the data recorded during the course of routine clinical practice (statistical data, case histories, surgery, and procedure reports) were taken from the electronic health record of the hospital by the first author. The following clinical and statistical data were included in the analysis: basic patient characteristics, tumor histology, EVD placement, surgical treatment, and shunt placement. An independent neuroradiologist reviewed preoperative MRI scans for the location of the tumor, tumor size, and preoperative hydrocephalus. Assessment of hydrocephalus was made by an Evans index (the maximum width between the frontal horns divided by the maximal width of the inner table) larger than 0.3 with or without clinical symptoms and signs (headache, nausea, vomiting, lethargy, papilledema, etc.) [16, 19]. Location of the tumor was classified as follows: superiorly (into the aqueduct of Sylvius), caudally (into the foramen magnum), laterally (into the foramen of Luschka or cerebellopontine angle), or anteriorly (invading or distorting the brainstem) (Fig 1). The extent of resection was evaluated by reviewing the postoperative MRI images obtained within 72 hours after surgical treatment, and the gross-total resection was defined as complete tumor resection with no evidence of residual tumor on postoperative MRI [20]. The number and duration of the external ventricular drain (EVD) and ventriculoperitoneal (VP) shunts were

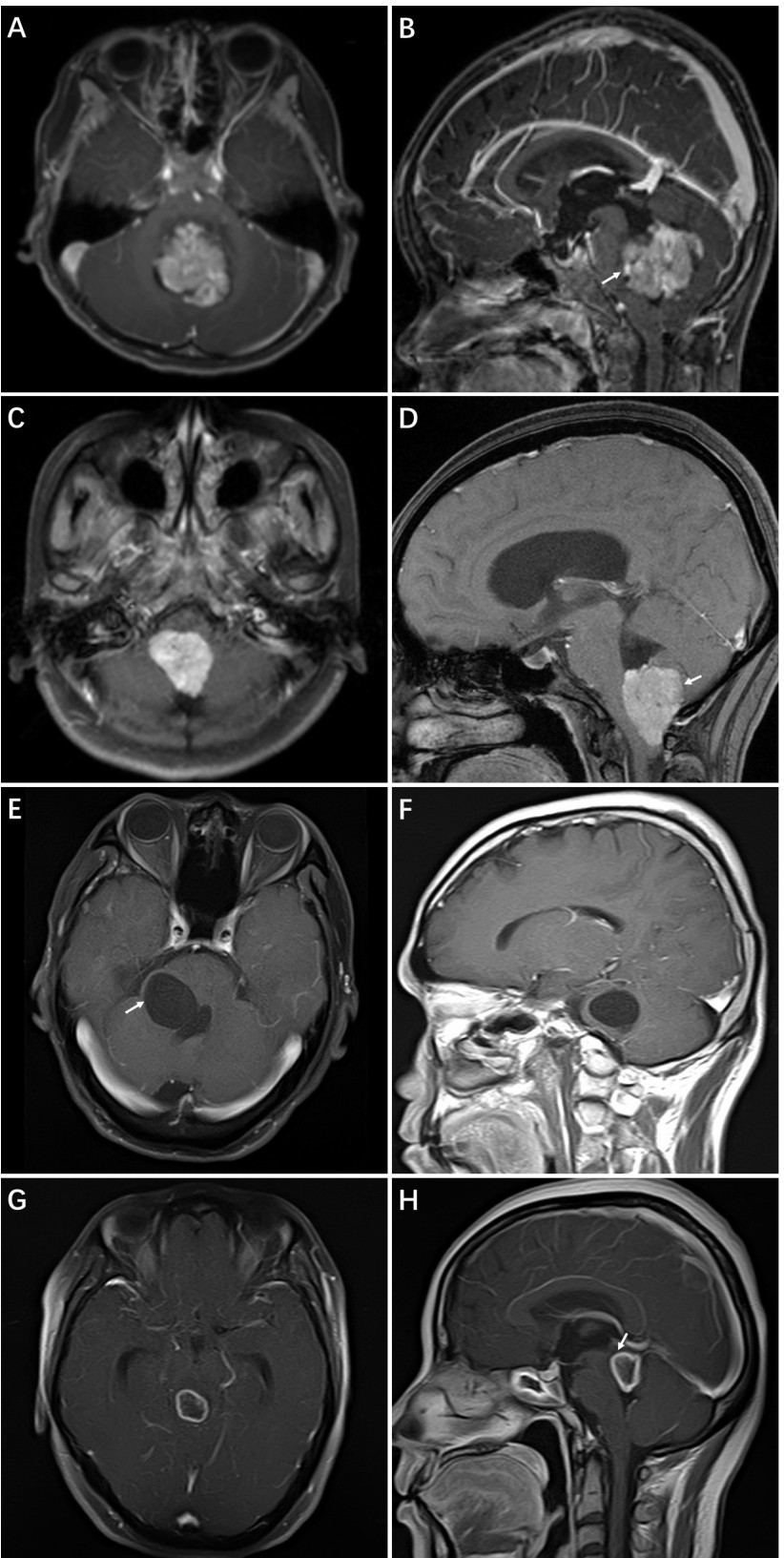

**Fig 1. Cases of extension of fourth ventricle tumor.** (**A** and **B**) The axial and sagittal contrast-enhanced T1-weighted image shows the lesion invading into the brainstem. (**C** and **D**) The axial and sagittal contrast-enhanced T1-weighted image shows the lesion invading into the foramen magnum. (**E** and **F**) The axial and sagittal contrast-enhanced T1-weighted image shows the lesion invading into the left foramen of Luschka. (**G** and **H**) The axial and sagittal contrast-enhanced T1-weighted image shows the lesion invading into the aqueduct of Sylvius.

noted after surgery. The follow-up was done on postoperative days 30 and 90 and at 6 months after surgery.

## Clinical management and surgical procedures

The surgical procedures were all performed by three senior neurosurgeons (Yuekang Z, Xuhui H, and Yan J). Resection of the tumor was performed with the patient in a prone or a park bench/side position. The lesions were approached through a standard midline suboccipital approach. The indication of prophylactic EVD placement included one of the following two types: 1) symptomatic hydrocephalus and 2) asymptomatic ventriculomegaly. Prophylactic EVD placement was performed immediately after preoperative hydrocephalus had been confirmed by imaging radiological examination. Gross-total removal or subtotal resection was achieved in all patients. During the operation, electrophysiological monitoring was performed. Postoperatively, all patients were monitored in the ICU for at least 1 day. In patients with EVD, CSF was drained via the EVD for 3–5 days. The EVD was removed when no further symptomatic elevated ICP appeared over a period of turning off EVD for 24 hours. In patients without EVD placement, an EVD was rapidly placed if symptomatic high ICP was observed.

## Outcomes

The indication for VP shunt placement was a failed reduction of CSF drainage due to constantly elevated ICP with clinical symptoms (headache, vomiting, lethargy, papilledema, and upward gaze paresis) lasting for more than 1–2 weeks. The postoperative hydrocephalus was defined as symptomatic hydrocephalus requiring CSF drainage following tumor removal.

## Statistical analysis

Data analysis was performed using the SPSS software version 23.0 (IBM Corp., Armonk, New York, USA). Categorical variables were reported as counts (%) and continuous variables were described by median (interquartile range [IQR]). The chi-square test and Fisher exact test were employed to complete the univariate analysis of patients with and without postoperative hydrocephalus. Differences in continuous variables between patients with and without postoperative hydrocephalus were compared by the Wilcoxon-Mann-Whitney test. The significant factors in the univariate analysis were used as covariates in the multivariate analysis, which was performed using logistic regression. All statistical tests were 2-sided; if P <0.05, the data were considered statistically significant.

## Results

### Patient characteristic

In total, 121 patients (60 males and 61 females) were included in our study. The median patient age was 24 years (IQR, 9–41 years). Histologically, the most common tumor was ependymoma (30.6%), followed by medulloblastoma (24.2%) and pilocytic astrocytoma (16.5%). Rare lesions included cholesteatoma, hemangioblastoma, vascular malformation, choroid plexus papilloma, and metastatic lesions. Most tumors extended beyond the boundaries of the fourth

**Table 1. Patients' characteristics and details of tumor.**

| Variables | Value |
|---|---|
| Sex | |
| Female | 61 (50.4%) |
| Male | 60 (49.6%) |
| Tumor size (mm) | 37 (30–44) |
| Age (years) | 24 (9–41) |
| <3 | 11 (9.1%) |
| 3–5 | 10 (8.3%) |
| 5–18 | 35 (28.9%) |
| >18 | 65 (53.7%) |
| Tumor pathology | |
| Ependymoma | 37 (30.6%) |
| Medulloblastoma | 29 (24.0%) |
| Astrocytoma | 20 (16.5%) |
| Hemangioblastoma | 7 (5.8%) |
| Cholesteatoma | 7 (5.8%) |
| Choroid plexus papilloma | 7 (5.8%) |
| Metastatic | 4 (3.3%) |
| Non-Hodgkin lymphoma | 1 (0.8%) |
| Other | 9 (7.5%) |
| Tumor growth characteristics [a] | |
| Lateral extension | 34 (28.1%) |
| Anterior extension | 98 (81.0%) |
| Caudal extension | 69 (57.0%) |
| Superior extension | 14 (11.6%) |
| No extension beyond the fourth ventricle | 8 (6.6%) |

Values are number of patients (%) or median (interquartile range).

[a] Percentages do not add up to 100 because some patients had more than 1 growth characteristic.

ventricle. The anterior extension was the most frequent and was observed in 81.0% of cases, while the caudal extension (57.0%) was the least frequent. The lateral extension was apparent in 28.1% of cases. The superior extension was found in 11.6% of cases. In terms of surgery, gross-total resection was achieved in 91 patients (73.4%), and there was no perioperative mortality. Details are shown in **Table 1**.

## Predictors for postoperative hydrocephalus which need CSF diversion

Overall, there were 56 patients with prophylactic EVD placement and 65 patients without EVD insert before tumor resection. Fifteen patients underwent postoperative CSF diversion, which included 10 VP shunts and 5 EVDs. Of the 10 VP shunts, 9 cases had a prophylactic EVD placement, and only one underwent postoperative EVD before VP shunt placement. This accounted for 17.9% (10/56) of the patients with prophylactic inserted EVDs. In the univariate analysis, the need for postoperative CSF drainage in patients was significantly correlated with sex (p = 0.012), superior extension (p = 0.015), preoperative hydrocephalus (p<0.001), and subtotal resection (p<0.001) (**Table 2**). Meanwhile, the need for postoperative VP in patients was significantly correlated with superior extension (p = 0.016), preoperative hydrocephalus (p<0.005), prophylactic EVD (0.006), and subtotal resection (p<0.001)

**Table 2. Univariate analysis of the association between each factor and postoperative hydrocephalus.**

| Variables | Postoperative CSF diversion | | P-value |
|---|---|---|---|
| | Yes (15) | No (106) | |
| Sex | | | 0.012[a] |
| Female | 3 (4.9%) | 51 (95.1%) | |
| Male | 12 (20%) | 48 (80%) | |
| Tumor size (mm) | 40 (34–49) | 36 (30–43) | 0.100 |
| Age (years) | 18 (3–38) | 24 (9–41) | 0.301 |
| Tumor pathology | | | |
| Ependymoma | 3 (8.1%) | 34 (91.9%) | 0.226[b c] |
| Medulloblastoma | 5 (17.2%) | 24 (82.8%) | 1.0[b c] |
| Astrocytoma | 4 (20.0%) | 16 (80.0%) | |
| Lateral extension | | | 0.553[b] |
| Yes | 3 (8.8%) | 31 (91.2%) | |
| No | 12 (13.8%) | 75 (86.2%) | |
| Anterior extension | | | 0.159[b] |
| Yes | 88 (89.8%) | 10 (10.2%) | |
| No | 5 (21.7%) | 18 (78.3%) | |
| Caudal extension | | | 0.420[a] |
| Yes | 10 (14.5%) | 59 (85.5%) | |
| No | 5 (9.6%) | 47 (90.4%) | |
| Superior extension | | | 0.015[b] |
| Yes | 5 (35.7%) | 9 (64.3%) | |
| No | 10 (9.3%) | 97 (90.7%) | |
| Extent of resection | | | <0.001[b] |
| GTR | 3 (3.3%) | 87(96.7%) | |
| STR | 12 (38.7%) | 19(61.3%) | |
| Preoperative hydrocephalus | | | <0.001[a] |
| Yes | 15 (21.4%) | 55 (78.6%) | |
| No | 0 (0%) | 51 (100%) | |
| Prophylactic EVD | | | 0.091[a] |
| Yes | 10 (17.9%) | 46 (82.1%) | |
| No | 5 (7.7%) | 60 (92.3%) | |

CSF, cerebrospinal fluid; GTR, gross total resection; STR, subtotal resection; EVD, external ventricular drainage.

[a] Chi-square test.

[b] Fisher exact test.

[c] p value compared with astrocytoma.

(**Table 3**). In the multivariate analysis, superior extension (p = 0.013; OR = 44.761; 95% CI 2.235–896.310) and subtotal resection (p = 0.005; OR = 0.087; 95% CI 0.016–0.473) were confirmed as independent influencing factors for CSF drainage requirement (**Table 4**). However, only prophylactic EVD (p = 0.042; OR = 10.431; 95% CI 1.093–99.514) were confirmed as independent influence factors for postoperative VP. When considering the subgroup of patients who had prophylactic EVD placement prior to surgery, superior extension (p = 0.003), and subtotal resection (p<0.001) were significantly correlated with the need for postoperative CSF drainage (**Table 5**). Multivariate analysis identified superior extension (p = 0.012; OR = 23.400; 95% CI 1.982–276.230) and subtotal resection (p = 0.004; OR = 0.036; 95% CI 0.004–0.344) as the independent influencing factors (**Table 6**).

**Table 3. Univariate analysis of the association between each factor and postoperative VP.**

| Variables | Postoperative VP | | P-value |
|---|---|---|---|
| | Yes (10) | No (110) | |
| Sex | | | 0.054[b] |
| Female | 2 (3.3%) | 59 (96.7%) | |
| Male | 8 (13.3%) | 52 (86.7%) | |
| Tumor size (mm) | 44 (33–53) | 36 (30–43) | 0.032 |
| Age (years) | 16 (1–45) | 24 (9–41) | 0.533 |
| Tumor pathology | | | |
| Ependymoma | 2 (5.4%) | 35 (94.6%) | 0.170[b c] |
| Medulloblastoma | 3 (10.3%) | 26 (89.7%) | 0.422[b c] |
| Astrocytoma | 4 (20.0%) | 16 (80.0%) | |
| Lateral extension | | | 0.724[b] |
| Yes | 2 (5.9%) | 32 (94.1%) | |
| No | 8 (9.2%) | 79 (90.8%) | |
| Anterior extension | | | 0.339[b] |
| Yes | 7 (7.1%) | 91 (92.9%) | |
| No | 3 (13%) | 20 (87%) | |
| Caudal extension | | | 1.000[b] |
| Yes | 6 (8.7%) | 63 (91.3%) | |
| No | 4 (7.7%) | 48 (92.3%) | |
| Superior extension | | | 0.016[b] |
| Yes | 4 (28.6%) | 10 (71.4%) | |
| No | 6 (5.6%) | 101 (94.4%) | |
| Extent of resection | | | <0.001[b] |
| GTR | 1 (1.1%) | 89(98.9%) | |
| STR | 9 (29%) | 22(71%) | |
| Preoperative hydrocephalus | | | <0.005[b] |
| Yes | 10 (14.3%) | 60 (85.7%) | |
| No | 0 (0%) | 51 (100%) | |
| Prophylactic EVD | | | 0.006[b] |
| Yes | 9 (16.1%) | 47 (83.9%) | |
| No | 1 (1.5%) | 64 (98.5%) | |

[a] Chi-square test.

[b] Fisher exact test.

[c] p value compared with astrocytoma.

## Discussion

There is still no consensus for management of hydrocephalus before and after fourth ventricle lesions surgery. In the present study, we retrospectively analyzed clinical parameters, including sex, age, preoperative hydrocephalus, tumor type, size and location, the extent of resection, and prophylactic EVD to identify whether these factors were correlated with persistent

**Table 4. Multivariate analysis of factors associated with postoperative hydrocephalus.**

| Variables | Odds ratio (95% CI) | p-value |
|---|---|---|
| Superior extension | 44.761(2.235–896.310) | 0.013 |
| Gross total resection | 0.087(0.016–0.473) | 0.005 |

**Table 5. Univariate analysis of the association between each factor and postoperative hydrocephalus in a subgroup of perioperative EVD placement.**

| Variables | Postoperative CSF diversion | | p-value |
|---|---|---|---|
| | Yes (10) | No (46) | |
| Sex | | | 0.162[b] |
| Female | 2(8.3%) | 22(91.7%) | |
| Male | 8(25.0%) | 24(75.0%) | |
| Tumor size (mm) | 41(33–50) | 36(31–41) | 0.174 |
| Age (years) | 22.5(1–45) | 25.5(11–41) | 0.528 |
| Tumor pathology | | | |
| Ependymoma | 3(20.0%) | 12(80.0%) | 0.239[b c] |
| Medulloblastoma | 2(15.4%) | 11(84.6%) | 0.357[b c] |
| Astrocytoma | 4(36.4%) | 7(63.6%) | |
| Lateral extension | | | 0.713[b] |
| Yes | 2(13.3%) | 13(86.7%) | |
| No | 8(19.5%) | 33(80.5%) | |
| Anterior extension | | | 0.361[b] |
| Yes | 7(15.2%) | 39(84.8%) | |
| No | 3(30.0%) | 7(70.0%) | |
| Caudal extension | | | 0.727[b] |
| Yes | 7(20.0%) | 28(80.0%) | |
| No | 3(14.3%) | 18(85.7%) | |
| Superior extension | | | 0.003[b] |
| Yes | 5(62.5%) | 3(37.5%) | |
| No | 5(10.4%) | 43(89.6%) | |
| Extent of resection | | | <0.001[b] |
| GTR | 2(5.0%) | 38(95.0%) | |
| STR | 8(50.0%) | 8(50.0%) | |

[a] Chi-square test.

[b] Fisher exact test.

[c] p value compared with astrocytoma.

postoperative hydrocephalus. Superior tumor extension and failed total resection of tumors were identified as significant risk factors for the development of postoperative hydrocephalus. To the best of our knowledge, this is the first study confirming that superior tumor extension (into the aqueduct) increases the incidence of postoperative hydrocephalus. Our findings may be used to preoperatively identify patients at high risk of postoperative hydrocephalus after resection of the fourth ventricle tumor.

Previous studies have shown that younger patients with posterior fossa are at higher risk for the development of persistent hydrocephalus.[3, 4, 15, 21, 22] Bognar et al.[3] and Kumar et al. [21] have demonstrated that age< 3 years at tumor surgery is a significant predictor of postoperative CSF diversion. This might be explained by the finding that the younger patient with

**Table 6. Multivariate analysis of factors associated with postoperative hydrocephalus in a subgroup of perioperative EVD placement.**

| Variables | Odds ratio (95% CI) | p-value |
|---|---|---|
| Superior extension | 23.400(1.982–276.230) | 0.012 |
| Gross total resection | 0.036(0.004–0.344) | 0.004 |

fourth ventricle tumor related to the higher incidence of congenital and malignant tumors, like medulloblastoma, frequently accompanied by leptomeningeal metastases which may develop impaired CSF absorption at the subarachnoid level because of the more aggressive nature of tumors [22–24]. However, we did not find age to be significantly associated with postoperative CSF drainage, which might be explained by a larger number of enrolled adults in the current compared to previous studies and a small group of medulloblastoma. The relationship between age and postoperative hydrocephalus in patients with fourth ventricle tumor needs to be further explored.

There are conflicting findings on the association between preoperative hydrocephalus and postresection hydrocephalus. Our results did not confirm preoperative hydrocephalus to be correlated with the need for a postoperative CSF drainage. However, Gopalakrishnan et al. [7] and Morelli et al. [18] found that patients with severe hydrocephalus on preoperative MRI were at a higher rate of postoperative CSF division procedure, which might be because severe hydrocephalus leads to higher venous and CSF pressures and possibly a longer time to resolution of the pressure. This inconsistency with previous reports may be due to evaluating preoperative hydrocephalus by a dichotomized variable rather than calculating the extent of hydrocephalus into 3 groups (mild, moderate, and marked).

In previous studies, tumors in posterior fossa predominantly located in the midline were associated with a higher incidence of postoperative CSF diversion procedure compared with those tumors situated in the cerebellar hemispheres [12, 15, 25]. In our study, we classified the location of tumors by displayed extension: superior extension (into the aqueduct), caudal extension (into the foramen magnum), lateral extension (into the foramen of Luschka or cerebellopontine angle), and anterior extension (invading or distorting the brainstem). Notably, we find superior extension was a significant predictor for needing the postoperative CSF division, while other extensions were not associated with a postoperative shunt. The inflation after resection of tumor could result in stenosis of the CSF pathway in patients with tumor extension (aqueduct, foramina of Luschka, and foramen of Magendie). For the management of hydrocephalus in patients with fourth ventricle tumor, endoscopic third ventriculostomy (ETV) is a safe and durable means of controlling obstruction hydrocephalus and the risk of ETV failure may be lower than the risk of shunt failure surgery when ETV Success Score (ETVSS) $\geq$ 80 [26–30]. However, fourth ventricle tumor resection frequently developed adhesive arachnoiditis and secondary hydrocephalus when some cases with communication hydrocephalus after tumor resection could benefit from shunt rather than ETV in the study. In cases where the CSF flow from the aqueduct was well seen, there was a trend for shunt insertion based on the assumption that there was impaired CSF absorption at the subarachnoid level. Factors like the presence of postoperative cerebellar edema, especially in cases with triventricular hydrocephalus appeared on imaging, favored ETV rather than shunt insertion.

In the present study, the size of the tumor was not correlated with the need for a postoperative CSF drainage. Similarly, in their study, Sherise et al. [16] found no association between the size of the tumor and persistent postoperative hydrocephalus. The study by Kumar et al. [21] demonstrated that incidence of a CSF diversion procedure was highest after the surgical removal of medulloblastoma and ependymoma, and lowest among patients with astrocytoma, which may be explained by the observed higher incidence of astrocytoma in a lateral rather than a midline location and the higher number of such patients in the series. Meanwhile, Won et al. [23] found medulloblastoma was a predictor for the development of hydrocephalus, including noncommunicating hydrocephalus due to occlusion of the fourth ventricle or its outlets and communicating hydrocephalus secondary to leptomeningeal metastases due to the more aggressive nature of medulloblastoma. Yet, no significant correlation was found in this study. Inconsistency between our results and those reported by previous studies may be

explained by the small group of medulloblastoma. The small group of each kind of tumor in this study might make us fail to find the association between pathology and the incidence of postoperative hydrocephalus.

In the present series, gross total resection of the tumor was statistically associated with a lower incidence of the need for postoperative CSF diversion procedure, by the way, there is no residual tumor near the aqueduct and the residual tumors are located in the brain stem and the bottom of the fourth ventricle in failed-total resection cases on postoperative MRI. This association was also confirmed by Culley et al. [15], Kumar et al. [21], and Gnanalingham et al. [31], which may be due to residual tumor obstructing CSF flow. Conversely, more recent studies did not find a relationship between the extent of tumor resection and postoperative hydrocephalus [2, 5, 7, 32], which could be due to the less volume of the residual tumor with sophisticated surgical techniques and the continuous development of neurosurgical instruments. However, compared with GTR, whether near-total resection ($< 1.5 \text{ cm}^3$ residual) could lead to a higher incidence of postoperative hydrocephalus remained unknown because the volume of residual tumor was not accurately measured in these studies. Therefore, further studies are necessary to determine the association between the volume of residual tumor and postoperative hydrocephalus.

Our results showed no statistically significant association between the prophylactic EVD placement and postoperative CSF drainage. In contrast to our study, Culley et al. [15] demonstrated that a more extended EVD placement might be significantly correlated with persistent hydrocephalus. A possible explanation for this may be a different indication of prophylactic EVD placement in our study. Nevertheless, it is important to note that prolonged placement of an EVD might be a sign of difficult weaning instead of a physiological predictor of persistent hydrocephalus.

The present study has some limitations. First, as a single-center retrospective study, admission bias may be present in our sample. Second, details of the treatment of posterior fossa tumor, including surgical techniques and perioperative management, may vary among hospitals. Thus, our findings should be confirmed by other multi-center prospective studies with a larger sample.

## Conclusion

Superior tumor extension (into the aqueduct) and failed total resection of tumor resulted as significant risk factors for postoperative hydrocephalus in patients with fourth ventricle tumor. These findings may help to identify the patients who are at high risk of postoperative hydrocephalus and require medical interventions.

## Supporting information

**S1 Table. Patients' characteristics and details of tumor.**
(PDF)

**S2 Table. Univariate analysis of the association between each factor and postoperative hydrocephalus.**
(PDF)

**S3 Table. Univariate analysis of the association between each factor and postoperative VP.**
(PDF)

**S4 Table. Multivariate analysis of factors associated with postoperative hydrocephalus.**
(PDF)

**S5 Table. Univariate analysis of the association between each factor and postoperative hydrocephalus in a subgroup of perioperative EVD placement.**
(PDF)

**S6 Table. Multivariate analysis of factors associated with postoperative hydrocephalus in a subgroup of perioperative EVD placement.**
(PDF)

**S1 File. STROBE Statement.**
(DOCX)

## Author Contributions

**Conceptualization:** Yanming Ren.

**Formal analysis:** Bowen Huang, Zhigang Lan.

**Funding acquisition:** Yanming Ren, Yuekang Zhang.

**Investigation:** Tengyun Chen.

**Methodology:** Tengyun Chen, Chenghong Wang, Wenke Liu.

**Project administration:** Tengyun Chen, Yuekang Zhang.

**Resources:** Yan Ju, Xuhui Hui, Yuekang Zhang.

**Software:** Bowen Huang.

**Supervision:** Yanming Ren.

**Writing – original draft:** Tengyun Chen.

**Writing – review & editing:** Yanming Ren.

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
