## [Decision Letter · Decision Letter 0]

7 Sep 2020

PONE-D-20-23644

Risk factors for hydrocephalus following fourth ventricle tumor surgery: A retrospective analysis of 121 patients

PLOS ONE

Dear Dr. Zhang,

Thank you for submitting your manuscript to PLOS ONE. After careful consideration, we feel that it has merit but does not fully meet PLOS ONE’s publication criteria as it currently stands. Therefore, we invite you to submit a revised version of the manuscript that addresses the points raised during the review process.

The reviewers did have rather divergent opinions on the manuscript. Please adress the issues brought forward by the reviewers, and change the manuscript where reasonable. Where this is not possible, please explain your standpoint.

We look forward to receiving your revised manuscript.

Kind regards,

Michael C Burger, M.D.

Academic Editor

PLOS ONE

Journal Requirements:

Reviewers' comments:

Reviewer's Responses to Questions

**Comments to the Author**

1. Is the manuscript technically sound, and do the data support the conclusions?

Reviewer #1: Yes

Reviewer #2: Partly

Reviewer #3: Yes

Reviewer #4: Yes

Reviewer #5: No

Reviewer #6: Yes

Reviewer #7: Yes

2. Has the statistical analysis been performed appropriately and rigorously? 

Reviewer #1: Yes

Reviewer #2: Yes

Reviewer #3: Yes

Reviewer #4: Yes

Reviewer #5: I Don't Know

Reviewer #6: Yes

Reviewer #7: Yes

3. Have the authors made all data underlying the findings in their manuscript fully available?

Reviewer #1: No

Reviewer #2: Yes

Reviewer #3: Yes

Reviewer #4: Yes

Reviewer #5: Yes

Reviewer #6: Yes

Reviewer #7: Yes

4. Is the manuscript presented in an intelligible fashion and written in standard English?

Reviewer #1: Yes

Reviewer #2: Yes

Reviewer #3: Yes

Reviewer #4: Yes

Reviewer #5: Yes

Reviewer #6: Yes

Reviewer #7: Yes

5. Review Comments to the Author

Reviewer #1: This study is a retrospective chart review that addresses the incidence of persistent postoperative hydrocephalus requiring surgery, in patients undergoing surgery for resection of fourth ventricular tumors. The study adds significantly to existing knowledge about this issue. The study includes a large number of patients, who were treated using a relatively standardized protocol. The analysis was thoughtfully designed and uses statistics appropriately. I believe that this study will be useful for practicing neurosurgeons, in order tocounsel patients and their families appropriately about the likelihood of needing additional surgeries (ETV or shunt) to treat hydrocephalus, after resection of 4th ventricular tumors, and in order to assess the likelihood of EVD weaning after these surgeries.

Reviewer #2: PONE-D-20-23644

Title: Risk factors for hydrocephalus following fourth ventricle tumor surgery: A retrospective analysis of 121 patients

Aim: Identify risk factors for postoperative hydrocephalus and the need for a postoperative CSF diversion.

Retrospective study of 121 patients with fourth ventricle tumor treated with tumor resection (histologically verified single intracranial lesion). Main outcome variable – post resection CSF diversion (EVD or VP-shunt)

Main results: 121 patients. Median age was 24 years (IQR, 9-41 years). The most common tumors were ependymoma (30.6%), followed by medulloblastoma (24.2%) and pilocytic astrocytoma (16.5%). 56/121 patients had prophylactic EVD placement. Fifteen patients underwent postoperative CSF diversion, which included 10 VP shunts and 5 EVDs. Of the 10 VP shunts, 9 cases had a prophylactic EVD placement, and only one underwent postoperative EVD before VP shunt placement. Multivariate analysis identified superior extension and subtotal resection as independent risk factors for post resection CSF drainage (EVD or VP-shunt).

Comments:

1. The manuscript is well written and easy to read.

2. The topic is relevant and of general interest.

3. My main objection to this manuscript is the choice of main outcome variable. In my opinion, the combination of post-resection EVD and permanent VP is confusing, all the time 56/121 had placement of EVD before resection. I would have preferred to use permanent VP alone as main outcome variable. Doing this, pre-operative HC will most likely remain significantly associated with post-resection VP placement. I encourage the authors to at least include such an analysis. It is of course interesting to note that 5/65 patients without pre-resection placement of EVD needed EVD in the early postoperative period, but only on of these five ended up with VP.

4. Identifying superior extension and subtotal resection as independent risk factors for post resection CSF drainage (EVD or VP-shunt) is absolutely of clinical interest, but will be even more interesting if analyzed according to the comment above.

5. Pre-resection endoscopic third ventriculostomy (ETV) is an alternative to EVD, and may reduce the need for permanent VP. The mention of ETV in the discussion is somewhat brief.

Reviewer #3: The authors have sufficiently addressed my concerns and feel that with the changes the article is not suitable for publication.

The authors have significantly enhanced the discussion section that was previously lacking.

Reviewer #4: This is a nice series which outlines the 4th ventricular tumor situations which are more likely to lead to persistent hydrocephalus requiring shunting. The authors are to be congratulated on this analysis.

Reviewer #5: This is a large series but the results are not surprising. The conclusions are simple yet well known to the reader. Several comments that deserve further discussion are

1. what were indications for preoperative EVD placement and timing for such procedure in relation to tumor surgery.

2. What was the timing for definitive CSF diversion surgery in relation to the index tumor operation

3. What was incidence of complications with the shunt procedure

4. What was the shunt infection rate?

5. Did the authors consider an ETV rather than VP shunt

Reviewer #6: Manuscript review: Risk factors for hydrocephalus following fourth metrical tumor surgery: a retrospective analysis of 121 patients. (PONE-D-20-23644)

The authors of this retrospective review describe their experience from 2013-2018 with 121 patients who presented with fourth ventricular tumors and had a first time resection surgery for the lesion. Patients had a median age of 24 years and all had six months follow up after initial surgery.

The objective of the study was to identify factors that predispose such patients to develop hydrocephalus and needing (ventriculo-peritoneal shunt) VPS for CSF diversion. Of the total 121 patients enrolled 16 patients eventually underwent permanent VPS placement. The univariate analysis demonstrated that superior extension into the aqueduct, preoperative hydrocephalus, and subtotal resection of the mass were significantly associated with postoperative hydrocephalus. This retrospective analysis is well written and a significant univariate and multivariate statistical analysis was completed in an attempt to identify specific factors associated with this patient population but a few questions arise.

The median age of the patient population is 24 years of age and this should be more explicitly mentioned, perhaps even in the title of the manuscript, as typically fourth ventricular tumors tend to be more highly concentrated in a pediatric population.

The conclusions from the paper state that superior tumor extension and failed gross total resection were independent risk factors for VPS placement. However this makes one wonder whether a subtotal resection within the aqueduct led to hydrocephalus. If this is the case then this is an obvious conclusion in that the obstruction through the ventricular system was not alleviated. However, if indeed, superior extension and failed total resection were not factors related to the same patients then this needs to be further explained in the manuscript and the manuscript would benefit from the authors’ hypotheses on how this would then lead to hydrocephalus. In addition to having a data analysis on initial extension of the tumor on patient presentation, a further explanation of which areas of tumor were left behind when a gross total resection was not possible would also assist in explaining the results.

Further, the authors mentioned that endoscopic third ventriculostomy (ETV) was a consideration for patients with fourth ventricular obstruction. The paper does not address why none of the patients who eventually underwent VPS placement were or were not considered for ETV as this may have been able to avoid permanent CSF diversion through a shunt.

In conclusion the manuscript nicely evaluates many different factors associated with this patient population however the manuscript in its current state seems to address the conclusions in a rather simplistic fashion and as such the conclusions do not seem to add much new knowledge in the field.

Reviewer #7: The authors present a series of 121 patient's with posterior fossa tumors to try to determine prognostic indicators for the development of postoperative hydrocephalus that requires CSF diversion. There is a basic assumption in this manuscript which may be falacious: Regardless of pathology it is the anatomic extent of the tumor that determines hydrocephalus. Although using appropriate statistical analysis the office did not demonstrate significance of pathology, the subgroup's may be too small to provide a true "real world answer".

My 2nd concern is that this is predominantly an article of teenagers and young adults. In the discussion the role of age and its implication for development of hydrocephalus is briefly reviewed but I believe this needs to be further expanded. It is not just age but the more aggressive nature of the tumors that we see in children under 5 years of age and particularly under 3 years of age with leptomeningeal dissemination that may increase the risk of hydrocephalus. The older patients in this study and the relatively small number of those with medulloblastoma may give a false impression of watch factors truly influence the development of hydrocephalus.

The statistically significant finding of superior extent of the tumor increasing the risk for postoperative hydrocephalus is novel and of interest. I would be curious if this finding would hold up if the authors limited the population studied 2 patients over 16 years of age, that is those that are anatomically adult. I am not certain that this finding would be seen in younger children, particular with medulloblastoma where the mean age is below that of the study population in this manuscript.

6. PLOS authors have the option to publish the peer review history of their article (what does this mean?). If published, this will include your full peer review and any attached files.

Reviewer #1: **Yes: **Ronald Benveniste

Reviewer #2: **Yes: **Eirik Helseth

Reviewer #3: No

Reviewer #4: **Yes: **Lawrence M Shuer

Reviewer #5: No

Reviewer #6: No

Reviewer #7: No

---

## [Author Response · Author response to Decision Letter 0]

24 Sep 2020

Dear Editors:

Thank you for your letter and comments concerning our manuscript entitled "Risk factors for hydrocephalus following fourth ventricle tumor surgery: A retrospective analysis of 121 patients" (PONE-D-20-23644). We have studied the comments carefully and made corresponding modifications. All modifications in the manuscript were highlighted in red. The main corrections in the paper and the responses to the reviewers’ and editor’s comments are as follows:________________________________________

Reviewers’ comments:

Reviewer #1

1. This study is a retrospective chart review that addresses the incidence of persistent postoperative hydrocephalus requiring surgery, in patients undergoing surgery for resection of fourth ventricular tumors. The study adds significantly to existing knowledge about this issue. The study includes a large number of patients, who were treated using a relatively standardized protocol. The analysis was thoughtfully designed and uses statistics appropriately. I believe that this study will be useful for practicing neurosurgeons, in order to counsel patients and their families appropriately about the likelihood of needing additional surgeries (ETV or shunt) to treat hydrocephalus, after resection of 4th ventricular tumors, and in order to assess the likelihood of EVD weaning after these surgeries.

Authors’ response: Thank you for the comments.

Reviewer #2

1. The manuscript is well written and easy to read.

Authors’ response: Thank you for the comment.

2. The topic is relevant and of general interest.

Authors’ response: Thank you for the comment.

3. My main objection to this manuscript is the choice of main outcome variable. In my opinion, the combination of post-resection EVD and permanent VP is confusing, all the time 56/121 had placement of EVD before resection. I would have preferred to use permanent VP alone as main outcome variable. Doing this, pre-operative HC will most likely remain significantly associated with post-resection VP placement. I encourage the authors to at least include such an analysis. It is of course interesting to note that 5/65 patients without pre-resection placement of EVD needed EVD in the early postoperative period, but only one of these five ended up with VP.

Authors’ response: We have added this analysis. (Page8, line172-179) 

4. Identifying superior extension and subtotal resection as independent risk factors for post resection CSF drainage (EVD or VP-shunt) is absolutely of clinical interest, but will be even more interesting if analyzed according to the comment above.

Authors’ response: It has been analyzed according to the comment above.

5. Pre-resection endoscopic third ventriculostomy (ETV) is an alternative to EVD, and may reduce the need for permanent VP. The mention of ETV in the discussion is somewhat brief.

Authors’ response: We have added some details about EVT in the discussion. (Page13, line 264-274) 

Reviewer #3

1. The authors have sufficiently addressed my concerns and feel that with the changes the article is not suitable for publication. The authors have significantly enhanced the discussion section that was previously lacking.

Authors’ response: Thank you for your comment. But we did not make a revision before.

Reviewer #4

1. This is a nice series which outlines the 4th ventricular tumor situations which are more likely to lead to persistent hydrocephalus requiring shunting. The authors are to be congratulated on this analysis.

Authors’ response: Thank you for the comment.

Reviewer #5

1. what were indications for preoperative EVD placement and timing for such procedure in relation to tumor surgery 

Authors’ response: We have described the indications in the sentence (Page5, line 114-116). And we have added details about the timing for preoperative EVD placement (Page5, line 116-118). 

2. What was the timing for definitive CSF diversion surgery in relation to the index tumor operation

Authors’ response: We have described them in the sentence (Page 5, line 126-128).

3. What was incidence of complications with the shunt procedure

Authors’ response: No serious complications occurred in any of the cases of VP shunting during the follow-up period.

4. What was the shunt infection rate?

Authors’ response: No patient with shunt infection was found during the follow-up period. 

5. Did the authors consider an ETV rather than VP shunt

Authors’ response: Thank you for your valuable comment. We did not routinely perform ETV when the patients in this study were treated for hydrocephalus. But now, ETV was performed more than VP in our hospital. Moreover, the type of surgery for postoperative hydrocephalus do not affect the occurrence of postoperative hydrocephalus as well as the risk factors.

Reviewer #6

1. The median age of the patient population is 24 years of age and this should be more explicitly mentioned, perhaps even in the title of the manuscript, as typically fourth ventricular tumors tend to be more highly concentrated in a pediatric population.

Authors’ response: We have explicitly mentioned and added details about age in the outcome. (Table 1) 

2. The conclusions from the paper state that superior tumor extension and failed gross total resection were independent risk factors for VPS placement. However, this makes one wonder whether a subtotal resection within the aqueduct led to hydrocephalus. If this is the case then this is an obvious conclusion in that the obstruction through the ventricular system was not alleviated. However, if indeed, superior extension and failed total resection were not factors related to the same patients then this needs to be further explained in the manuscript and the manuscript would benefit from the authors’ hypotheses on how this would then lead to hydrocephalus. In addition to having a data analysis on initial extension of the tumor on patient presentation, a further explanation of which areas of tumor were left behind when a gross total resection was not possible would also assist in explaining the results.

Authors’ response: I am sorry we did not mention this information. There were no cases with the residual tumor in the aqueduct. During the operation, the surgeons ensured regarding the adequacy of visualization of the aqueduct and CSF flow after tumor removal. Meanwhile, there is no residual tumor near the aqueduct and the residual tumors are located in the brain stem and the bottom of the fourth ventricle in failed-total resection cases on postoperative MRI. We have added this sentence (Page 14, line 296-299).

3. Further, the authors mentioned that endoscopic third ventriculostomy (ETV) was a consideration for patients with fourth ventricular obstruction. The paper does not address why none of the patients who eventually underwent VPS placement were or were not considered for ETV as this may have been able to avoid permanent CSF diversion through a shunt.

Authors’ response: We do not routinely use ETV for hydrocephalus with fourth ventricle tumor, and this will be improved in the future. Thank you for this valuable comment.

Reviewer #7

1. There is a basic assumption in this manuscript which may be fallacious: Regardless of pathology it is the anatomic extent of the tumor that determines hydrocephalus. Although using appropriate statistical analysis the office did not demonstrate significance of pathology, the subgroup's may be too small to provide a true "real world answer".

Authors’ response: Thank you for this valuable comment. We have mentioned this in the limitations. (Page, line 282-294)

2. In the discussion the role of age and its implication for development of hydrocephalus is briefly reviewed but I believe this needs to be further expanded. It is not just age but the more aggressive nature of the tumors that we see in children under 5 years of age and particularly under 3 years of age with leptomeningeal dissemination that may increase the risk of hydrocephalus. The older patients in this study and the relatively small number of those with medulloblastoma may give a false impression of watch factors truly influence the development of hydrocephalus.

Authors’ response: We have further expanded the part in the discussion (Page 13, line 230-240). Thank you for this valuable comment.

3. I would be curious if this finding would hold up if the authors limited the population studied 2 patients over 16 years of age, that is those that are anatomically adult. I am not certain that this finding would be seen in younger children, particular with medulloblastoma where the mean age is below that of the study population in this manuscript.

Authors response: Superior tumor extension and subtotal resection were identified as independent risk factors for hydrocephalus after resection of tumor in the fourth ventricle in this study. But only subtotal resection was significantly associated with hydrocephalus when only patients older than 16 years were enrolled. This seems to be a type II error, which might be attributed to a smaller sample size when pediatric patients were excluded.

Finally, we earnestly appreciate the Editor’s helpful work and hope that these answers and corrections will meet with approval. Once again, thank you very much for your comments and suggestions. 

Best regards.

Yuekang Zhang

Department of Neurosurgery, West China Hospital, Sichuan University, China

E-mail: 2012zykyx@sina.cn

---

## [Decision Letter · Decision Letter 1]

12 Oct 2020

PONE-D-20-23644R1

Risk factors for hydrocephalus following fourth ventricle tumor surgery: A retrospective analysis of 121 patients

PLOS ONE

Dear Dr. Zhang,

Thank you for submitting your manuscript to PLOS ONE. After careful consideration, we feel that it has merit but does not fully meet PLOS ONE’s publication criteria as it currently stands. Therefore, we invite you to submit a revised version of the manuscript that addresses the points raised during the review process.

Please add a paragraph into the discussion part of the manuscript where you explain whether or not less than 1.5 cc of residual tumor coating the floor of the fourth ventricle will necessitate ultimate CSF diversion.

We look forward to receiving your revised manuscript.

Kind regards,

Michael C Burger, M.D.

Academic Editor

PLOS ONE

Reviewers' comments:

Reviewer's Responses to Questions

**Comments to the Author**

1. If the authors have adequately addressed your comments raised in a previous round of review and you feel that this manuscript is now acceptable for publication, you may indicate that here to bypass the “Comments to the Author” section, enter your conflict of interest statement in the “Confidential to Editor” section, and submit your "Accept" recommendation.

Reviewer #1: All comments have been addressed

Reviewer #2: All comments have been addressed

Reviewer #4: All comments have been addressed

Reviewer #5: All comments have been addressed

Reviewer #7: (No Response)

2. Is the manuscript technically sound, and do the data support the conclusions?

Reviewer #1: Yes

Reviewer #2: Yes

Reviewer #4: Yes

Reviewer #5: Yes

Reviewer #7: Partly

3. Has the statistical analysis been performed appropriately and rigorously? 

Reviewer #1: Yes

Reviewer #2: Yes

Reviewer #4: Yes

Reviewer #5: I Don't Know

Reviewer #7: Yes

4. Have the authors made all data underlying the findings in their manuscript fully available?

Reviewer #1: No

Reviewer #2: Yes

Reviewer #4: Yes

Reviewer #5: No

Reviewer #7: Yes

5. Is the manuscript presented in an intelligible fashion and written in standard English?

Reviewer #1: Yes

Reviewer #2: Yes

Reviewer #4: Yes

Reviewer #5: Yes

Reviewer #7: Yes

6. Review Comments to the Author

Reviewer #1: I recommend publication in its current form

I recommend publication in its current form

I recommend publication in its current form

Reviewer #2: In my opinion, the authors have responded well to the reviewers comments, and the revised manuscript is improved compared to the original manuscript

Reviewer #4: I believe the revisions improve the quality of the article.

I believe the authors have answered the reviewers concerns to my satisfaction.

Reviewer #5: (No Response)

Reviewer #7: I still take issue with the comment that a gross total resection versus a near total resection is a significant indicator for the need of a shunt procedure particularly in younger children. I would urge the authors to further expand the discussion, particularly questioning whether less than 1.5 cc of residual tumor coating the floor of the fourth ventricle will necessitate ultimate CSF diversion. This is particularly pertinent since it has been well-established in the pediatric neuro-oncology literature that there is no survival advantage for gross total resection versus minimal residual disease

7. PLOS authors have the option to publish the peer review history of their article (what does this mean?). If published, this will include your full peer review and any attached files.

Reviewer #1: **Yes: **Ronald Benveniste

Reviewer #2: **Yes: **Eirik Helseth

Reviewer #4: **Yes: **Lawrence M. Shuer, MD

Reviewer #5: No

Reviewer #7: No

---

## [Author Response · Author response to Decision Letter 1]

20 Oct 2020

Editor’s comments：

1. Please add a paragraph into the discussion part of the manuscript where you explain whether or not less than 1.5 cc of residual tumor coating the floor of the fourth ventricle will necessitate ultimate CSF diversion.

Authors’ response: We have further expanded the part in the discussion (Page 14, line 288-298). Thank you for this valuable comment.

Reviewers’ comments:

Reviewer #1

1. I recommend publication in its current form

Authors’ response: Thank you for the comments.

Reviewer #2

1. In my opinion, the authors have responded well to the reviewers comments, and the revised manuscript is improved compared to the original manuscript.

Authors’ response: Thank you for the comment.

Reviewer #4

1. I believe the revisions improve the quality of the article. I believe the authors have answered the reviewers concerns to my satisfaction.

Authors’ response: Thank you for the comment.

Reviewer #5

(No Response)

Reviewer #7

1. I still take issue with the comment that a gross total resection versus a near total resection is a significant indicator for the need of a shunt procedure particularly in younger children. I would urge the authors to further expand the discussion, particularly questioning whether less than 1.5 cc of residual tumor coating the floor of the fourth ventricle will necessitate ultimate CSF diversion. This is particularly pertinent since it has been well-established in the pediatric neuro-oncology literature that there is no survival advantage for gross total resection versus minimal residual disease.

Authors’ response: We have further expanded the part in the discussion (Page 14, line 288-298). Thank you for this valuable comment.

---

## [Editor Report · Decision Letter 2]

22 Oct 2020

Risk factors for hydrocephalus following fourth ventricle tumor surgery: A retrospective analysis of 121 patients

PONE-D-20-23644R2

Dear Dr. Zhang,

We’re pleased to inform you that your manuscript has been judged scientifically suitable for publication and will be formally accepted for publication once it meets all outstanding technical requirements.

Kind regards,

Michael C Burger, M.D.

Academic Editor

PLOS ONE
---

## [Editor Report · Acceptance letter]

9 Nov 2020

PONE-D-20-23644R2 

Risk factors for hydrocephalus following fourth ventricle tumor surgery: A retrospective analysis of 121 patients 

Dear Dr. Zhang:

I'm pleased to inform you that your manuscript has been deemed suitable for publication in PLOS ONE. Congratulations! Your manuscript is now with our production department. 

Kind regards, 

on behalf of

Dr. Michael C Burger 

Academic Editor

PLOS ONE